# Regulation of the Host Immune Microenvironment in Periodontitis and Periodontal Bone Remodeling

**DOI:** 10.3390/ijms24043158

**Published:** 2023-02-05

**Authors:** Nannan Han, Yitong Liu, Juan Du, Junji Xu, Lijia Guo, Yi Liu

**Affiliations:** 1Laboratory of Tissue Regeneration and Immunology and Department of Periodontics, Beijing Key Laboratory of Tooth Regeneration and Function Reconstruction, School of Stomatology, Capital Medical University, Beijing 100050, China; 2Immunology Research Center for Oral and Systemic Health, Beijing Friendship Hospital, Capital Medical University, Beijing 100050, China; 3Department of Orthodontics, School of Stomatology, Capital Medical University, Beijing 100050, China

**Keywords:** periodontitis, immune regulatory network, alveolar bone remodeling, immune cells

## Abstract

The periodontal immune microenvironment is a delicate regulatory system that involves a variety of host immune cells including neutrophils, macrophages, T cells, dendritic cells and mesenchymal stem cells. The dysfunction or overactivation of any kind of local cells, and eventually the imbalance of the entire molecular regulatory network, leads to periodontal inflammation and tissue destruction. In this review, the basic characteristics of various host cells in the periodontal immune microenvironment and the regulatory network mechanism of host cells involved in the pathogenesis of periodontitis and periodontal bone remodeling are summarized, with emphasis on the immune regulatory network that regulates the periodontal microenvironment and maintains a dynamic balance. Future strategies for the clinical treatment of periodontitis and periodontal tissue regeneration need to develop new targeted synergistic drugs and/or novel technologies to clarify the regulatory mechanism of the local microenvironment. This review aims to provide clues and a theoretical basis for future research in this field.

## 1. Introduction

Periodontitis is a chronic inflammatory disease caused by the dysbiosis of oral microbiota, which is the result of both host immune and inflammatory responses and is characterized by the integrity of damaged tooth-supporting tissues. At present, clinical nonsurgical treatment can control the infection, but these interventions are not enough to obtain long-term and stable therapeutic effects. Recent studies have confirmed that host immune cells play an important role in this process [1]. The periodontal immune microenvironment consists of a large diversity of cells, the extracellular matrix and various cytokines, which interact and create a sophisticated network. In the pathogenesis of periodontitis, the epithelial barrier of periodontal tissue, innate immune cells, IgA, IgG and IgM antibodies in gingival crevicular fluid and saliva constitute the first line of defense against the invasion and destruction of periodontal tissue by exogenous plaque. Once this line of defense is broken, disturbances of the periodontal immune microenvironment by pathogens lead to an increased inflammatory milieu, influence the differentiation and activity of host cells and finally lead to the degradation of periodontal supporting tissues [2]. Studies have indicated that bacteria, such as *Porphyromonas gingivalis*, *Tannerella forsythia* (*forsythensis*) and *Treponema denticola*, might be the main pathogenic microbiota of periodontitis occurrence [3,4]. However, with the development of research on the mechanism of periodontal disease, an increasing number of studies hold the opinion that periodontitis is not caused by only one or even several kinds of pathogenic microorganisms but is also closely related to an imbalance in the periodontal microenvironment after bacterial infections in periodontal tissues [5]. These microbiota coordinatively cause dysbiosis of the oral microecosystem in periodontitis-susceptible hosts [6,7]. Some pathogenic microorganisms, such as *P. gingivalis*, may not destroy periodontal tissue directly but can induce inflammation by secreting a large number of virulence factors, such as gingipains, endotoxins, organic acids and other metabolites, which play important roles in the coaggregation, biofilm formation and oral microbial dysbiosis of periodontal tissues [8]. It is known that more than 300,700 kinds of bacterial species exist in the human oral cavity [9]. Normally, most microbiota are permanent residents of the local microenvironment, and a dynamic equilibrium exists between those bacteria and the innate host defense system. The planktonic bacteria involved can be removed, while the pathogens attached to the biofilm are difficult to remove using the current nonsurgical treatment methods [6,10,11]. To date, we have concluded that periodontitis should be considered a disease associated with dental plaque biofilms that are formed on the surface of teeth. The structural and functional heterogeneity of biofilms allows microbiota to adhere well to the tooth surface, avoid the host immune system and inhibit chemotaxis and the functions of immune cells and stem cells [12], which leads to tissue inflammation and destruction.

The development of oral dysbiosis may last for a period of time, gradually changing the symbiotic host–microbe relationship to a pathogenic relationship, which is the primary microbial factor contributing to periodontitis, while the primary immunological factor is the destructive host inflammatory response. The gingival epithelium not only acts as a physical barrier to microorganisms in the host immune defense but also recognizes microbiota-associated molecular patterns (MAMPs) by expressing various pattern recognition receptors (PRRs) and reacts with MAMPs by secreting various cytokines and chemical factors such as IL-8 and antibacterial peptides, which plays an active biological barrier role in host immune recognition and initiation and actively participates in the host’s innate immune response and acquired immune response [13,14,15]. Despite the difficulty in defining the mechanisms of periodontitis, it is certain that the immune status of the local periodontal microenvironment in healthy tissues and periodontitis tissues are different, in which several types of host immune cells such as macrophages, T cells and host stem cells are involved [16] (shown schematically in Figure 1). Therefore, it is necessary to consider changes in the periodontal microenvironment and host immune regulation in the process of periodontal regeneration.

## 2. Bilateral Regulation of Neutrophils in Periodontitis

It is currently recognized that the main pathogenic factors of periodontitis are gram-negative bacteria, including *P. gingivalis* and *T. forsythia*, which are mainly colonized in periodontal pockets [3,4,10]. These bacterial pathogenic infections stimulate and disturb the homeostasis of the local immune microenvironment, which causes gingivitis, and the inflammatory reaction of gingivitis triggers and maintains the inflammatory reaction of oral microorganisms. If the inflammation of the gingiva is not effectively controlled, it will cause periodontitis, eventually resulting in an imbalance between local osteogenesis and osteoclastic functions, which causes the destruction and resorption of the alveolar bone [17]. As nonspecific pathogen killer cells, neutrophils are essential in the pathogenesis of periodontitis and play an important role in immune defense responses. With further research on neutrophil biology, an increasing number of studies have concluded that periodontitis is a disease model dominated by neutrophils [18] and that the degree of periodontal tissue damage is highly related to the function of local neutrophils [19].

Neutrophils are the most important phagocytes and are the most prevalent immune cells in the human body. Neutrophils in the gingival sulcus are the first line of defense against periodontal pathogens. Mature neutrophils are stimulated by bacteria and its product LPS, and under the regulation of cytokines, adhesion molecules and chemokines, which pass through vascular endothelium through a series of activities such as adhesion and chemotaxis, can reach the inflammatory site to engulf bacteria and then kill bacteria by releasing lysosomal enzymes or respiratory bursts [20,21]. Neutrophils are not only an important defense cell in the process of periodontitis but also have a dual role in causing inflammation [22]. If neutrophils react violently to pathogenic stimuli they can cause immune damage to the body [23]. Neutrophils have the ability of protein synthesis and participate in the induction of the immune response by synthesizing and releasing cytokines with immunomodulatory effects [24]. Neutrophils in peripheral blood and the gingival sulcus can synthesize or secrete cytokines such as IL-1, TNF-α, IL-6 and IL-8 and prostaglandin E2 (PGE2), which cause the aggravation and expansion of inflammation [25]. Previous studies have indicated that the immune regulatory functions of neutrophils are necessary for periodontal health, and the phagocytic function and killing ability of neutrophils in periodontitis tissues are impaired. However, in recent years, through a clinical study of samples from periodontitis patients, it has been found that neutrophils in periodontal tissues infiltrate in large amounts, and their survival time is significantly prolonged, which results in a significant correlation between the number of local excessive neutrophils and the destruction of periodontal tissue [23]. Neutrophils isolated from clinically inflamed periodontal tissues also show excessive activity characteristics [26]. These results indicate that the functions of neutrophils play a two-way role in the pathogenesis of periodontitis. The phagocytic killing function of neutrophils under physiological conditions is critical for periodontal homeostasis; however, hyperactive neutrophils amplify inflammation and aggravate tissue damage. The pathogen clearance functions of neutrophils involve a series of mechanisms, including cell adhesion and chemotaxis, phagocytosis, respiratory bursts and the formation of neutrophil extracellular traps (NETs) [27]. Neutrophil respiratory bursts are one of the most important mechanisms for killing pathogenic microorganisms. After chemotaxis into the periodontal tissue and the phagocytosis of pathogens, neutrophils activate the respiratory burst pathway under the action of the chemokine CXCL8/CXCR1 signal axis [28].

Nicotinamide adenine dinucleotide phosphate (NADPH) oxygenase is rapidly assembled from two membrane protein subunits, p22phox (CYBA) and gp91phox (CYBB), which specifically bind to the cytoplasmic subunits Ncf1 and Ncf2 to activate the downstream mitogen-activated protein kinase (MAPK) pathway, producing large amounts of nitric oxide (NO) and reactive oxygen species (ROS). NO is an important cell-signaling factor and is a molecule that kills infecting microorganisms. ROS are the main weapons for neutrophil sterilization [29,30]. During this process, NADPH oxidase reduces oxygen molecules (O_2_) to superoxide anions (O^−2^), and simultaneously, the oxygen consumption of cells increases significantly. Such a physiological behavior is called a respiratory burst. Activated neutrophils can release large amounts of NO and ROS through respiratory bursts, killing antigens and bacteria. However, excessive NO accumulation amplifies local inflammatory reactions, and excessive ROS production also leads to the oxidative stress of local cells and causes periodontal tissue damage [18].

NETs are a type of extracellular fibrous reticular structure produced by neutrophils after stimulation by pathogens, which are mainly composed of macromolecular polymers of unzipped chromatin and antimicrobial-related complexes such as histones and antimicrobial peptides. Electron microscopic observations have shown that NETs are based on scaffolds composed of a DNA histone complex with a diameter of 15–17 nm, embedded in a spherical structure with a maximum diameter of 50 nm, that includes neutrophil elastase, myeloperoxidase and citrullinated histone H3 [31]. The process by which neutrophils are stimulated to produce NETs is called NETosis, which can be divided into suicidal NETosis and vital NETosis [32,33]. Suicidal NETosis is a form of NET accompanied by neutrophil death, which is mainly manifested by nuclear deconcentration, cell membrane disintegration and self-DNA excretion to the outside of the cell, expressing a fibrous reticular structure [34]. The respiratory bursts of neutrophils and ROS have been shown to play key roles in the process of inducing suicidal NETosis, and this activation pathway is closely related to the MAPK signaling pathway [35]. Vital NETosis can be further subdivided into nuclear DNA-releasing NETosis and mitochondrial DNA-releasing NETosis. In the process of nuclear DNA-releasing NETosis, the cell membrane and the nuclear membrane remain intact, and the nuclear DNA is released in the form of vesicles through budding, while the phagocytic function of neutrophils is not affected. This process does not depend on the MAPK pathway or on ROS production [36]. The DNA released by mitochondrial DNA-releasing NETosis comes from intracellular mitochondria, and its formation depends on the ROS pathway, which also ensures the integrity and function of the remaining cells and does not accompany the death of neutrophils [37].

Multiple studies have shown that NETs play an active antibacterial role in periodontal tissue. However, excessive NETs increase the expression of inflammatory cytokines and the degree of inflammation in the local microenvironment, resulting in harmful effects and periodontal destruction. Clinical studies have shown that the expression level of NETs in periodontal tissues and in the gingival crevicular fluid of periodontal patients is significantly higher compared with healthy controls [38,39], which can reduce the level of local NETs, thus, inhibiting periodontal inflammation and promoting tissue regeneration [40]. Moreover, studies have confirmed that ascorbic acid (vitamin C) derived from vegetables and fruits is involved in collagen hydroxylation as a cofactor of many enzymes, which can prevent oxidative damage to DNA and intracellular proteins. In plasma, vitamin C can increase endothelium-dependent vasodilation and reduce extracellular oxidants produced by neutrophils, thus, reducing the inflammation of periodontitis [41,42]. In addition, Litcubanine A (LA), which is derived from the traditional Chinese medicinal plant, Litsea cubeba, can reduce periodontitis bone loss by inhibiting the expression levels of the neutrophil respiratory burst-related and inflammation-related genes CYBB and NCF2 [43]. These studies indicate that the formation of NETs by neutrophils and proinflammatory mechanisms may become new targets and strategies for the treatment of periodontitis. However, the effect of NETs on the development of periodontitis and the mechanism of alveolar bone destruction and resorption requires further investigation.

## 3. Damage and Repair Functions of Macrophages

When tissues are damaged or are in an inflammatory state, macrophages play a central role in mobilizing the host defense mechanism, which participates in the recognition, phagocytosis, presentation and elimination of these antigens as the first line of defense of the immune system. For decades, there has been a consensus that at least two different phenotypes of macrophages can be distinguished: classically activated macrophages (M1) and alternatively activated macrophages (M2), both of which are key factors in the progression of tissue inflammation and regeneration. Lipopolysaccharide (LPS) produced by *P. gingivalis* is a classic activator of M1 macrophages, while M2 macrophages are produced when cocultured with IL-4 and IL-13 in vitro [44,45]. Recent studies have confirmed that macrophages are involved in the processes of periodontitis [46,47]. At the early stage of inflammation, M1 macrophages are predominant and produce proinflammatory factors, such as IL-1, tumor necrosis factor (TNF), IL-6, ROS and NO, thus, enhancing the inflammatory response. At the later stage of the inflammatory reaction, M2 macrophages release an abundance of IL-10, anti-inflammatory cytokines and trophic factors, including transforming growth factor (TGF)-β and Arginase-1 (Arg1), to play an anti-inflammatory role and promote the repair of damaged tissues. The bacterial products in periodontal tissues lead to chemotaxis and the activation of macrophages in the local tissue. When inflammatory M1 macrophages are inhibited, the progression of periodontitis is also controlled, which further proves the correlation between macrophages and periodontitis [48,49,50].

The receptor activator of NF-κB ligand (RANKL)/receptor activator of NF-κB (RANK)/osteoprotegerin (OPG) axis plays a key role in the process of bone remodeling. Studies have found that RANKL mRNA expression is mainly located in inflammatory cells, including lymphocytes and macrophages, in periodontal tissues [51]. Moreover, a large number of M1 macrophage markers, such as IL-1 and TNF-α, have been detected in the bone destruction area of periodontal tissues [52,53]. It has also been found that large numbers of M1 macrophages infiltrate periodontitis tissues, while the number of M2 macrophages is small, which further indicates that M1 macrophages play a leading role in the occurrence and development of periodontitis [54]. All evidence indicates that M1 macrophages occupy the main position in periodontal tissues, thus, we conjectured that an imbalance between M1 and M2 macrophages enhances local inflammation, eventually leading to alveolar bone resorption. It has been found that the regeneration effect of alveolar bone was obviously improved due to the activation of M1 macrophages that was inhibited by aspirin (acetylsalicylic acid, ASA) treatment during wound healing [33]. All evidence suggests that tissue repair and regeneration induced by the host body itself are limited after eliminating inflammation, and macrophages are the key factor in “switching to the tissue repair model”. Using the tissue repair program detector, macrophages guide tissue regeneration by activating different phenotypes [55,56]. Therefore, the regulation of macrophage phenotype must be considered during guided tissue regeneration through the application of biological agents. Although the phagocytosis of antigens initially activates macrophages, their function can be further enhanced by various factors such as IFN-γ secreted by activated T cells. Macrophages and T cells interact during immune responses, which facilitates the activation of macrophages [57,58].

In recent years, many drugs have been developed to treat periodontitis via the polarization of macrophages. For example, Pelargonium dendritic cell root extract (PSRE) can inhibit IL-8 and PGE2 produced by fibroblasts induced by LPS and IL-6 produced by leukocytes and can inhibit CD80 and CD86 expression by macrophages, as well as the expression of IL-1 and COX-2 by leukocytes [59]. In addition, CSINCpi-2, metformin and C-C motif chemokine ligand 2 (CCL2) microparticles (MPs) can alleviate inflammation and inhibit alveolar bone loss by inhibiting the expression of inflammatory factors produced by macrophages [60,61,62]. However, the mechanisms of drug actions on macrophages and phenotypic control still need more research.

## 4. The Immunomodulatory Effects of T Cells and T Cell Subsets

In addition to innate immune cells, the balance between helper T cells Th1 and Th2 also plays a key role in the progression of periodontitis. Th1 cells upregulate cytotoxic immune responses by releasing interleukin (IL)-12, interferon (IFN) and tumor necrosis factor (TNF) and participate in the processes of delayed allergy or inflammation. However, Th2 cells can enhance B cell proliferation and produce IgE and IgG1 antibodies by releasing cytokines such as IL-4, IL-5, IL-6 and IL-13, thus, inducing humoral immune responses and taking part in rapid allergic reactions. Normally, the functions of Th1 and Th2 cells are dynamic. The disruption of this balance by exogenous antigens leads to inflammation or immune diseases [63].

Some studies have found that alterations in the active and quiescent stages of periodontitis are closely related to the balance between Th1 and Th2 cell immune responses [64,65]. In the quiescent stage of periodontitis, Th1 cells play a leading role, and cytokines secreted by Th1 cells, such as IFN and IL-12, can upregulate cellular immune responses and inhibit bone resorption. However, in the active stage of periodontitis, Th2 cell immune responses become the main response, with large numbers of infiltrating bone marrow-dependent lymphocyte B cells and plasma cells, while IFN and IL-12 related to Th1 cells are decreased [66]. In the process of periodontal tissue regeneration, the key role of Th2 cells and methods to promote Th2 cells need to be considered.

In recent years, two other T cell subpopulations have been discovered: Th17 and Treg cells. Many studies have recognized that different populations of T cells may play different roles in the progression of periodontitis, and the Th1/Th2 cell balance theory is not applicable for all different stages of periodontitis or all types of immune responses. Th17 cells enhance the ability of neutrophils and their innate immunity to defend against extracellular bacteria by releasing IL-1, IL-8, IL-23R, IL-17A, IL-17F, IL-6, TNF, monocyte chemoattractant protein-1 and matrix metalloproteinase. Thus, Th17 cells exaggerate inflammatory reactions and promote tissue destruction or bone resorption [67,68]. The other new T cell subpopulation was found to be CD4+CD25+Treg cells, which express the unique transcription factor forkhead box (Fox) P3. CD4+CD25+FoxP3+Treg cells come from the natural Treg (nTreg) cells produced by the thymus and the adaptive Treg (aTreg) cells transformed from CD4+CD25-cells in peripheral blood [69]. Treg cells inhibit the proliferation and activation of other T cells, thereby, downregulating inflammatory responses by secreting IL-10 and TGF. Similar to the Th1/Th2 cell balance, Treg and Th17 cells also work in mutual regulation and cooperate with other immune cells during inflammation. In an inflammatory microenvironment, dendritic cells (DCs) present antigens and secrete IL-6, which may induce naive T cells to differentiate into Th17 cells and upregulate the inflammatory response [70,71]. Therefore, T cells play a key role in the development of periodontitis and periodontal regeneration.

At present, drugs targeting T cells mainly regulate the differentiation of T cells to reduce inflammation and control the progress of periodontitis. A prolyl hydroxylase (PHD) with hydrogel-formulated inhibitor, 1,4-dihydrophenonthrolin-4-one-3-carboxylic acid (1,4-DPCA/hydrogel), which can increase the CXCR4-dependent accumulation of Treg cells, decreases the expression of proinflammatory cytokine genes and promotes alveolar bone regeneration in an experimental periodontitis model in mice [72]. In addition, other drugs, such as curcumin and calcitriol, can inhibit the loss of alveolar bone by increasing the number of Treg cells, decreasing the number of Th17 cells and changing the proportion and function of Th cell subsets [73,74]. However, the function of the Treg and Th17 cell balance in the process of inflammatory diseases remains unclear and requires further investigation.

## 5. Dendritic Cells–T Cells Regulatory Network

As mentioned above, DCs and T cells play synergistic roles in the development of periodontitis. As antigen-presenting cells, DCs can enhance the periodontal tissue inflammatory response by upregulating Th1 and Th17 cell activation [75]. Since TGF can induce both Treg and Th17 cell activation, if there is no antigen stimulation, DCs will not produce IL-6 and naïve T cells will differentiate into CD4+CD25+Treg cells, which inhibits the function of T cells and prevents the occurrence of autoimmune diseases. In periodontal tissues of periodontitis, *P. gingivalis* can stimulate mature DCs derived from individuals with chronic periodontitis to secrete IL-12 and IFN-γ, which promotes Th1 cell responses and enhances inflammation, and Th1 cell activity is correlated with the number of mature DCs in periodontal tissue [76]. Moreover, oral infections with *P. gingivalis* in mice increase the number of DCs, which is positively correlated with Th17 cell activation [77]. Evidence indicates that DCs enhance periodontal inflammation by activating T cell subpopulations, while other studies have found that DCs may induce a protective response through the induction of Th2 lymphocytes and the Treg response [78]. It has also been confirmed that DCs are the key regulators of T cell fate. The local recruitment and programming of DC biomaterials can guide T cell responses. The combination of granulocyte-macrophage colony-stimulating factor (GM-CSF) and thymic stromal lymphopoietin (TSLP) can promote the migration of DCs in vitro. The intracutaneous injection of an alginate hydrogel-releasing GM-CSF can increase the number of DCs, and the local addition of TSLP can promote an increase in FOXP3+ Treg cells and then locally regulate the periodontal microenvironment by enriching and stimulating the tolerance response of DCs [79]. The ultimate goal is to prevent the pathogenic inflammation of periodontal disease. Therefore, the role of DCs in periodontitis still needs to be further explored, while there is a consensus that the function of the DC–T cell regulatory network balance plays a key role in periodontal disease.

## 6. Effects of Host Mesenchymal Stem Cells on Immune Cell Regulation

In the periodontal microenvironment, stem cells play a key role in regulating cell proliferation and differentiation, thus, influencing the function of mesenchymal stem cells (MSCs) in tissue repair and regeneration [80]. Interestingly, local MSCs also showed significant immunomodulatory functions in the local periodontal microenvironment. Periodontal ligament stem cells (PDLSCs) in periodontal tissues of periodontitis show an immunosuppressive effect on monocyte-macrophages and suppress T cell proliferation [81], which not only suppresses the local immune response but also protects against the proliferation of other pathogens and aggravates dysbacteriosis in the local microenvironment. Therefore, the inhibition of local inflammation and the reversal of host stem cell functions are key factors in the treatment of periodontitis.

MSCs, including PDLSCs, bone marrow stem cells (BMSCs), dental pulp MSCs (DPMSCs), stem cells from human exfoliated deciduous teeth (SHEDs), gingival mesenchymal stem/progenitor cells (GMSCs) and stem cells from the apical papilla (SCAPs), have been widely studied in periodontal regeneration owing to their abilities to differentiate into osteoblasts, fibroblasts and cementoblasts [82,83]. Studies have shown that hPDLSCs transplanted into the subcutaneous tissue of immunodeficient mice can differentiate into cementum and periodontal ligament-like structures in local tissues, which preliminarily demonstrates the ability of MSCs to reconstruct periodontal tissues and supports the theoretical possibility of periodontal regeneration. However, the regulatory mechanism of MSCs in an inflammatory microenvironment remains controversial. Some studies have found that the inflammatory cytokines IFN-γ, TNF-α and IL-1β can inhibit the osteogenic differentiation, immunoregulation and proliferation functions of MSCs [84,85,86], whereas other studies have shown that low levels of inflammatory stimulation, such as with the stimulation of LPS, the immunomodulatory properties of mBMSCs were increased. Moreover, the expression of collagen I and osteocalcin (OCN) in human PDLSCs were downregulated under Pg-LPS stimulation conditions, and calcium formation was decreased in mBMSCs by LPS at 1 μg/mL [87,88,89]. This indicated that an inflammatory microenvironment would inhibit the differentiation ability of local PDLSCs, which is absolutely not beneficial for periodontal tissue rebuilding and regeneration, even after bacterial removal. Therefore, the number and differentiation function of PDLSCs must be enhanced during periodontal regeneration.

It is worth noting that the regulatory effect of stem cells on other immune cells in the microenvironment cannot be ignored, especially in stem cell therapy. In recent decades, stem cell transplantation has been shown to have a significant effect on promoting periodontal tissue regeneration. However, some studies have suggested that transplanted MSCs promote tissue healing by regulating the periodontal immune microenvironment rather than by directly acting on the functions of MSCs [90]. The immunoregulatory functions of MSCs depend largely on the degree of the surrounding inflammation of periodontal tissues [91]. In a low-inflammatory microenvironment, MSCs promote immune responses by secreting cytokines and recruiting immune cells to local areas. However, when the local tissue suffers from inflammation, such as in periodontitis, MSCs transform from proinflammatory types to anti-inflammatory types to prevent overactivation of the immune response [92,93]. Some studies have shown that PDLSCs show low immunogenicity and produce significant immunosuppression through PGE2-induced T cell anergy [94]. Moreover, in the process of stem cell transplantation, PDLSCs promote the regeneration of periodontal tissues by inducing the activation of M2 macrophages [95]. In addition, stem cell therapy in a minipig model of periodontitis decreases humoral immunity by inhibiting the activation and proliferation of B cells through intercellular contact, mainly mediated by programmed cell death protein 1 (PD-1) and PD-L1. Interestingly, transplanted stem cells may also enhance B cell activity by secreting IL-6 [96,97]. Therefore, the current mechanism by which stem cell therapy promotes periodontal regeneration remains unclear.

Because the host immune response and the suppression of host stem cell function play major roles in disease progression, antimicrobial therapies and host modulation therapy should be used in combination with plaque removal treatment [98]. Simultaneously, considering the mechanism of paracrine immune regulation of replanted stem cells to promote periodontal regeneration, a method based on extracellular vesicles (EVs) seems to provide a new strategy for the treatment of periodontitis, which could overcome many clinical limitations of stem cell transplantation. As an ideal carrier, EVs have been proven to inhibit the activity of NF-κB induction by LPS to alleviate inflammation [99]. Moreover, microRNA-155-5p in EVs derived from PDLSCs can increase SIRT1 expression in CD4+ T cells, thus, alleviating the Th17/Treg cell imbalance and inflammatory response [100]. Therefore, the anti-inflammatory and immunosuppressive effects of EVs in different tissues can be used as an effective tool to treat chronic inflammation such as periodontitis. The bioactivities of the EVs suggest perspectives allowing to highlight the loss of periodontal homeostasis. However, the in vivo use of EVs is still in its beginnings. The development of biomaterials is currently a line of research concerning EVs for periodontal regeneration. In the future, clinical periodontitis treatments need to develop a synergistic means to target the host cell functions to achieve regeneration of the periodontal tissue complex, while this should be based on clarifying the microenvironment regulation mechanism of stem cell therapy.

## 7. Conclusions and Future Perspectives

The occurrence and development of periodontitis are the results of the interaction between microbiota and the host. The etiology of periodontitis involves both dental plaque biofilms and the host immune response. In this review, we summarized the host cell regulatory network mechanisms in the pathogenesis and development of periodontitis. For the treatment of periodontitis, controlling inflammation is the first step, including scaling, scraping and root planning, as well as adjuvant antibiotic treatment. Meanwhile, drug treatment is not the only option. Considering periodontitis as a bacterial infectious disease, we focused on the role of local cells in immune regulatory functions, including innate immune cells, adaptive immune cells and MSCs. The periodontal immune environment is a delicate regulatory system, in which the dysfunction or overactivation of any kind of local cell, and eventually an imbalance of the entire molecular regulatory network, can cause periodontal inflammation and tissue destruction (shown schematically in Figure 2). Therefore, the immune system is very efficient in preventing disease progression before the microbial dysbiotic environment has been established, and the treatment of periodontitis should focus not only on controlling the sources of infection and inflammation but also on remodeling the homeostasis of the host immune system in the future.

## Figures and Tables

**Figure 1 ijms-24-03158-f001:**
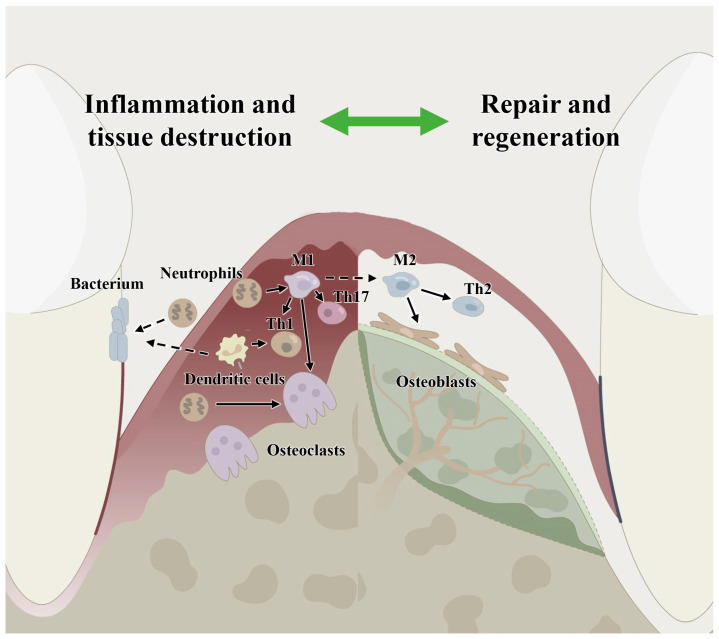
Host immune microenvironment regulation in periodontitis and in periodontal bone remodeling showing the involvement of immune cells, including macrophages, neutrophils, dendritic cells, T cells and host stem cells. M1: classical macrophages; M2: nonclassical macrophages; Th1: T helper cell 1; Th2: T helper cell 2; Th17: T helper cell 17.

**Figure 2 ijms-24-03158-f002:**
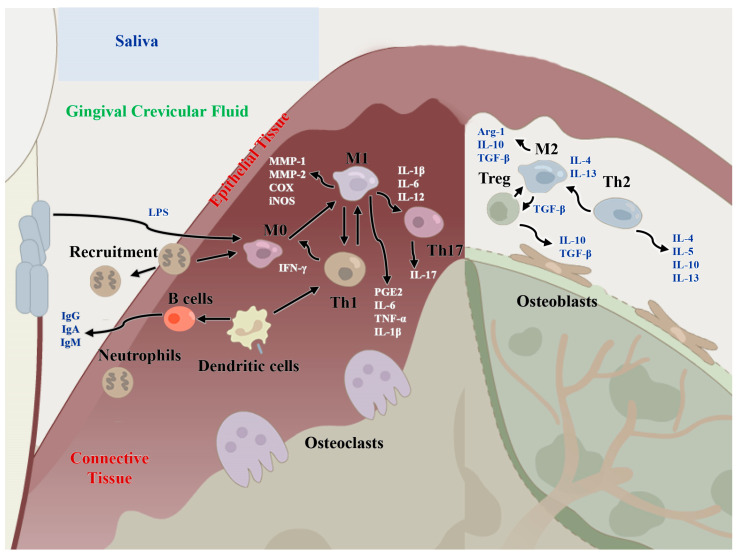
Regulatory functions of immune cells of the host immune microenvironment in periodontitis and periodontal bone remodeling. M1: classical macrophages; M2: nonclassical macrophages; Th1: T helper cell 1; Th2: T helper cell 2; Th17: T helper cell 17; MMP-1/2: matrix metalloproteinase 1/2; IL-: interleukin-, IFN-γ: interferon gamma; TNF-α: tumor necrosis factor alpha; TGF-β: transforming growth factor beta; Arg1: Arginase-1; PGE2: prostaglandin E2; COX: cyclooxygenase.

## Data Availability

The datasets used and analyzed in this study are available from the corresponding author (lililiuyi@163.com) upon reasonable request.

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
