# Peer review of "Regulation of the Host Immune Microenvironment in Periodontitis and Periodontal Bone Remodeling"

_ijms, 2023, doi:10.3390/ijms24043158_

Round 1
Reviewer 1 Report
This review provides a comprehensive and clear insight into the complex processes and events during the pathogenesis of periodontitis. I am sending several comments and suggestions with the intention to improve the quality of the text.
Line 29. Given that the destruction of periodontal ligament fibers is primarily a consequence of continuous inflammation and that bone resorption is not directly caused by microorganisms, I ask the authors to rephrase the sentence that says the deconstruction of the integrity of supporting tooth tissues is caused by pathogen invasion.
Line 36. As in the previous comment, I suggest rewording or avoiding the claim that microorganisms are the primary cause of tissue destruction. More relevant references should be used for the mentioned claims.
Line 45. Today, far more than 300 bacterial species are known in the human mouth, so I suggest to use current literature for this statement.
Line 48. Use the terms "biofilm" and “dental plaque” in the singular. It is unclear how biofilm is formed from different dental plaques. I suggest that this sentence should also be reworded to sound more common.

Author Response
Reviewer 1
This review provides a comprehensive and clear insight into the complex processes and events during the pathogenesis of periodontitis. I am sending several comments and suggestions with the intention to improve the quality of the text.
Line 29. Given that the loss of attachment and destruction of periodontal ligament fibers is primarily a consequence of continuous inflammation and that bone resorption is not directly caused by microorganisms, I ask the authors to rephrase the sentence that says the deconstruction of the integrity of supporting tooth tissues is caused by pathogen invasion.
RESPONSE: Thank you for your thorough review and salient observations. We have revised it in the Introduction section of the revised manuscript. Please see page 2 of the revised manuscript, lines 29 – 32, where we added the following text.
Periodontitis is a chronic inflammatory disease caused by the dysbiosis of oral microbiota, which is the result from both host immune and inflammatory responses, and is characterized by the integrity of tooth-supporting tissues damaged [1].
Reference:
1.Kwon, T.; Lamster, I.B.; Levin, L. Current concepts in the management of periodontitis. Int Dent J 2021, 71, 462-476.
Line 36. As in the previous comment, I suggest rewording or avoiding the claim that microorganisms are the primary cause of tissue destruction. More relevant references should be used for the mentioned claims.
RESPONSE: The authors appreciate this suggestion. We have corrected it in the revisedmanuscript. Please see page 2 of the revised manuscript, lines 46 – 47. where we added the following text.
Studies have indicated that bacteria, such as Porphyromonas gingivalis, Tannerella forsythia (forsythensis) and Treponema denticola, might be the main pathogenic microbiota of periodontitis occurrence [3,4].
References:
3. Hajishengallis, G.; Lamont, R. J.; Graves, D.T. The enduring importance of animal models in understanding periodontal disease. Virulence 2005, 6, 229-235.
4. Gemmell, E.; Yamazaki, K.; & Seymour, G. J. Destructive periodontitis lesions are determined by the nature of the lymphocytic response. Crit Rev Oral Biol Med 2002, 13,17-34.
Line 45. Today, far more than 300 bacterial species are known in the human mouth, so I suggest to use current literature for this statement.
RESPONSE: Thank you for this helpful suggestion. We have cited the latest reference in the revised manuscript. Please see page 11 of the revised manuscript, line 449.
Reference
9. Wade, W.G. The oral microbiome in health and disease. Pharmacol Res 2013, 69, 137-143.
Line 48. Use the terms "biofilm" and “dental plaque” in the singular. It is unclear how biofilm is formed from different dental plaques. I suggest that this sentence should also be reworded to sound more common.
RESPONSE: Thank you for your thorough review and salient observations. The authors have revised the text according to the Reviewer’s suggestion. Please see page 2 of the revised manuscript, lines 63 – 64, where we added the following text and reference.To date, we have concluded that periodontitis should be considered as a disease associated with dental plaque biofilms that are formed on the surface of teeth. The structural and functional heterogeneity of biofilms allows microbiota to adhere well to the tooth surface, avoid the host immune system, inhibit chemotaxis and the functions of immune cells and stem cells [12] , which leads to tissue inflammation and destruction.
Reference:
12. Balta, M.G.; Papathanasiou, E.; Blix, I.J.; Van Dyke, T.E. Host Modulation and
Treatment of Periodontal Disease. J Dent Res 2021, 100, 798-809.
Line 52. Bacterial migration from within gingival epithelial cells across an intact
basement membrane and into the gingival connective tissue has not been demonstrated in vivo. I ask the authors to reformulate or remove this part of the sentence.
RESPONSE: Thank you for your thorough review and helpful suggestion. We have deleted the sentence in the revised manuscript. Please see page 2 of the revised manuscript.
line 68.Figures 1 and 2. Below the figure, provide explanations for each abbreviation used in the figure.
RESPONSE: The authors appreciate this suggestion. We have corrected it in the revised manuscript.

Reviewer 2 Report
The authors wrote a review with the aim "to provide clues and a theoretical basis for future research". However, such a basis is not provided and the only suggestion for periodontitis treatment is to search for drugs to modulate the immune system. Nevertheless, no evidence is reported in the review for a dysfunctional immune response as the cause of periodontal disease.
In general, the writing is confusing. Periodontitis is characterized as an inflammatory disease (in the introduction) and as an infection (in the conclusions). Infections are the pathologic conditions that are caused by microbes. All infections stimulate inflammatory reactions. However, we do not call them inflammatory disorders. Inability of the defence system to remove the cause of the inflammation leads to the chronicity of the condition. This is one of the characteristics of biofilm-associated infections. Untreated chronic infections may cause tissue destruction, e.g. syphilis, tuberculosis, leprosy. This does not imply a dysfunctional immune system that has to be treated with drugs.
It is well-known that in most cases of adult periodontitis, biofilm control is enough to treat the infection. Is there any piece of evidence that the local immune response of these cases is different? Do the authors suggest new therapeutic strategies for the immune response in these cases?
There are many other confusing statements such as:
Lines 48-49: "To date, we have concluded that periodontitis should be considered as a disease associated with biofilms that are formed by various dental plaques." Dental plaque is a specific biofilm, the one formed on teeth, and not the constituent of the biofilm. I
Line 54: "The development of oral dysbiosis...". What is oral dysbiosis? How does it relate to periodontitis? Please, provide evidence.
Lines 69-70: "the main pathogenic factors of periodontitis are gram- negative bacteria, and their main component is red complex bacteria...". The specific plaque hypothesis (such as the one referring to the red complex) is no longer widely accepted as a valid one.
A new treatment for periodontitis is certainly of interest if it is a causal therapy. Do the authors imply that a therapeutic approach that affects the immune system can be a causal therapy? Certainly, periodontitis does not manifest in the absence of biofilm. Are the authors considering specific periodontitis forms that may benefit by applying a therapy for the immune system? Is there any evidence?
Author Response
Reviewer 2
1. The authors wrote a review with the aim "to provide clues and a theoretical basis for future research". However, such a basis is not provided and the only suggestion for periodontitis treatment is to search for drugs to modulate the immune system. Nevertheless, no evidence is reported in the review for a dysfunctional immune response as the cause of periodontal disease.
RESPONSE: The authors appreciate this constructive comment of the Reviewer. We have revised the text to address your concerns and hope that it is now clearer. Please see page 9 of the revised manuscript, lines 394 – 400, where we added the following text.
For the treatment of periodontitis, controlling inflammation is the first step, including scaling, scraping and root planning, as well as adjuvant antibiotic treatment. Meanwhile, drug treatment is not the only option [1,2]. Therefore, our aim is that the treatment of periodontitis should focus not only on controlling the sources of infection and inflammation, but also on remodeling the homeostasis of the host immune system in the future.
References:
1. Kwon, T.; Lamster, I.B.; Levin, L. Current concepts in the management of periodontitis.Int Dent J 2021, 71, 462-476.
2. Sedghi, L.M.; Bacino, M.; Kapila, Y.L. Periodontal Disease: The Good, The Bad, and The Unknown. Front Cell Infect Microbiol 2021, 11, 766944.
2. In general, the writing is confusing. Periodontitis is characterized as an inflammatory disease (in the introduction) and as an infection (in the conclusions). Infections are the pathologic conditions that are caused by microbes. All infections stimulate inflammatory reactions. However, we do not call them inflammatory disorders. Inability of the defence system to remove the cause of the inflammation leads to the chronicity of the condition. This is one of the characteristics of biofilm-associated infections. Untreated chronic infections may cause tissue destruction, e.g. syphilis, tuberculosis, leprosy. This does not imply a dysfunctional immune system that has to be treated with drugs.
RESPONSE: We greatly appreciate your supportive and helpful comments. We did notice a few sentences that were unclear because of some missing text. We have carefully rechecked and corrected all text in the revised manuscript. Please see page 1 of the revised manuscript, lines 30 – 36, where we added the following text.
Periodontitis is a chronic infectious disease. At present, the etiology involved in the study of periodontitis includes plaque microbial infection and host immune response. The clinical non-surgical treatment can control the infection, but these interventions are not enough to obtain long-term and stable therapeutic effects. Recent studies have confirmed that host immune cells play an important role in this process [1], so we summarized the preventive and therapeutic effects of periodontal host immune microenvironment in periodontitis.
Reference:
1. Kwon, T.; Lamster, I.B.; Levin, L. Current concepts in the management of periodontitis.Int Dent J 2021, 71, 462-476.
3. It is well-known that in most cases of adult periodontitis, biofilm control is enough to treat the infection. Is there any piece of evidence that the local immune response of these cases is different? Do the authors suggest new therapeutic strategies for the immune response in these cases?
RESPONSE: We greatly appreciate your supportive and helpful comments. We have revised the text to address your concerns and hope that it is now clearer. Please see page 9 of the revised manuscript, lines 394 – 400, where we added the following text.
Periodontitis is the result of the interaction between bacteria and the host. The etiology of periodontitis involves both dental plaque biofilm and host immune [1]. At present, mild to moderate cases can usually be managed by nonsurgical treatments, including auxiliary antibiotics, scaling, and root planning. However, the role of periodontal host immune microenvironment cannot be ignored [6,10,11]. In this review, we summarized the host cell regulatory network mechanisms in the pathogenesis, development and repair of periodontitis. We focused on the role of local cells in immune regulatory functions, including innate immune cells, adaptive immune cells and MSCs. Our goal is to provide
a new target for the prevention and treatment of periodontitis through immunomodulation.
References:
1. Kwon, T.; Lamster, I.B.; Levin, L. Current concepts in the management of periodontitis. Int Dent J 2021, 71, 462-476.
6. Curtis, M.A.; Diaz, P.I.; Van, T.E. The role of the microbiota in periodontal disease. Periodontol 2000 2020, 83, 14-25.
10. Sedghi, L.; DiMassa, V.; Harrington, A.; Lynch, S.V.; Kapila, Y.L. The oral
microbiome: Role of key organisms and complex networks in oral health and disease.Periodontol 2000 2021, 87, 107-131.
11. Li, W.; Zhang, Z.; Wang, Z. Differential immune cell infiltrations between healthy periodontal and chronic periodontitis tissues. BMC Oral Health 2020, 27, 293.
There are many other confusing statements such as:
4. Lines 48-49: "To date, we have concluded that periodontitis should be considered as a disease associated with biofilms that are formed by various dental plaques." Dental plaque is a specific biofilm, the one formed on teeth, and not the constituent of the biofilm.
RESPONSE: Thank you for your thorough review and salient observations. The authors have revised the text according to the Reviewer’s suggestion. Please see page 2 of the revised manuscript, lines 63 – 65, where we added the following text and reference. To date, we have concluded that periodontitis should be considered as a disease associated with dental plaque biofilms that are formed on the surface of teeth. by various dental plaques. The structural and functional heterogeneity of biofilms allows micro-biota to adhere well to the tooth surface, avoid the host immune system, inhibit chemotaxis and the functions of immune cells and stem cells [12], pass through the intact epithelial cell barrier and eventually invade the local tissue, which leads to tissue inflammation and destruction.
Reference:
12. Balta, M.G.; Papathanasiou, E.; Blix, I.J.; Van Dyke, T.E. Host modulation and
treatment of periodontal disease. J Dent Res 2021, 100, 798-809.
5. Line 54: "The development of oral dysbiosis...". What is oral dysbiosis? How does it relate to periodontitis? Please, provide evidence.
RESPONSE: Thank you for your thorough review and salient observations. The authors have revised the text according to the Reviewer’s suggestion. Please see page 2 of the revised manuscript, lines 51 – 52, where we added the following text and references. However, with the development of research on the mechanism of periodontal disease, an increasing number of studies hold the opinion that periodontitis is not caused by only one or even several kinds of pathogenic microorganisms but is also closely related to an imbalance of the periodontal microenvironment after bacterial infections in periodontal
tissues [6]. These microbiota coordinatively cause the dysbiosis of oral micro-ecosystem in periodontitis susceptible host [7].
References:
6. Curtis, M.A.; Diaz, P.I.; Van, T.E. The role of the microbiota in periodontal disease. Periodontol 2000 2020, 83, 14-25.
7. Lamont, R.J.; Koo, H.; Hajishengallis, G. The oral microbiota: dynamic communities and host interactions. Nat Rev Microbiol 2018, 16, 745-759.
6. Lines 69-70: "the main pathogenic factors of periodontitis are gram- negative bacteria, and their main component is red complex bacteria...". The specific plaque hypothesis (such as the one referring to the red complex) is no longer widely accepted as a validone.
RESPONSE: Thank you for this helpful suggestion. We have corrected it in the revised manuscript. Please see page 3 of the revised manuscript, line 94.
7. A new treatment for periodontitis is certainly of interest if it is a causal therapy. Do the authors imply that a therapeutic approach that affects the immune system can be a causal therapy? Certainly, periodontitis does not manifest in the absence of biofilm. Are the authors considering specific periodontitis forms that may benefit by applying a therapy for the immune system? Is there any evidence?
RESPONSE: We appreciate the Reviewer’s suggestion. Please see page 9 of the revised manuscript, lines 394 – 407 where we added the following text and references.
The occurrence and development of periodontitis is the result of the interaction between bacteria and the host. The etiology of periodontitis involves both dental plaque biofilm and host immune response. At present, mild to moderate cases can usually be managed by nonsurgical treatments, including auxiliary antibiotics, scaling, and root planning. Here, we mainly focus on the host immune microenvironment, expounding the functional changes of host immune cells during the occurrence and development of periodontitis, to provide new ideas for the treatment of periodontitis. In recent years, many studies have
confirmed the role of regulating immune microenvironment in the treatment of
periodontitis, including targeting neutrophils [40], macrophages [41,59], T cells [60,73], etc., as well as the methods of periodontal immunomodulation therapy using various cytokines and stem cells [91]. The most critical in periodontal disease pathogenesis is a reciprocally reinforced interplay between microbial dysbiosis and destructive inflammation. Pathogens induce periodontitis in susceptible patients and most of the time, the immune system is very
efficient to prevent disease progression before the microbial dysbiotic environment has been established.
References:
40. Van, V. U. Vitamin C and its role in periodontal diseases - The Past and the Present: A Narrative Review. Oral Health Prev Dent 2020, 18, 115-124.
41. Staudte, H.; Sigusch, B.W.; Glockmann, E. Grapefruit consumption improves vitamin C status in periodontitis patients. Br Dent J 2005, 199, 213-217.
59. Leguizamon, N.D.P.; Rodrigues, E.M.; Campos, M.L.; Nogueira, A.V.B.; Viola,
K.S.; Schneider, V.K.; et al. In vivo and in vitro anti-inflammatory and pro-osteogeniceffects of citrus cystatin CsinCPI-2. Cytokine 2019, 123, 154760.
60. Zhou, X.; Zhang, P.; Wang, Q.; J,i N.; Xia, S.; Ding, Y. Metformin ameliorates
experimental diabetic periodontitis independently of mammalian Target of rapamycin (mTOR) inhibition by reducing NIMA-related Kinase 7 (Nek7) expression. J Periodontol 2019, 90, 1032-1042.
73. Bi, C.S.; Li, X.; Qu, H.L.; Sun, L.J.; An, Y.; Hong, Y.L.; Tian, B.M.; Chen, F.M.
Calcitriol inhibits osteoclastogenesis in an inflammatory environment by changing the proportion and function of T helper cell subsets (Th2/Th17). Cell Prolif 2020, 53, e12827.
91. Gan, L.; Liu, Y.; Cui, D.; Pan, Y.; Zheng, L.; Wan, M. Dental tissue-derived human mesenchymal stem cells and their potential in therapeutic application. Stem Cells Int 2020,2020, 8864572.
Again, we appreciate your insightful comments and suggestions and we worked hard to respond to all of them. We have also made good use of the two articles you mentioned.Thank you for taking the time and energy to help us improve our paper.

Reviewer 3 Report
The immune factors of the host involved in periodontitis are explained in detail.
In the introduction, however, several errors must be addressed:
1- The invasion of bacteria in periodontitis is uncommon.
2- deconstruction of periodontal tissues (line 36). Please change to degradation
3 – Reference 5 is old. This number has climbed to 700 since 1998.
4- line 71: PG and TF are not IMPLANTED in periodontal pockets; instead, they COLONIZE them.
Please repair these minor errors.
Author Response
Reviewer 3
The immune factors of the host involved in periodontitis are explained in detail.
In the introduction, however, several errors must be addressed:
1. The invasion of bacteria in periodontitis is uncommon.
RESPONSE: Thank you for your thorough review and salient observations. We have revised it in the Introduction section of the revised manuscript. Please see page 2 of the revised manuscript, lines 29 – 32, where we added the following text.
Periodontitis is a chronic inflammatory disease caused by the dysbiosis of oral microbiota, which is the result from both host immune and inflammatory responses, and is characterized by the integrity of tooth-supporting tissues damaged [1].
Reference:
1. Kwon, T.; Lamster, I.B.; Levin, L. Current concepts in the management of periodontitis. Int Dent J 2021, 71, 462-476.
2. deconstruction of periodontal tissues (line 36). Please change to degradation.
RESPONSE: The authors recognize the Reviewer’s concern and have revised the
manuscript accordingly. Please see page 1 of the revised manuscript, line 43, where weadded the following text. Influence the differentiation and activity of host cells and finally lead to the degradation of periodontal supporting tissues [2].
Reference:
2. Sedghi, L.M.; Bacino, M.; Kapila, Y.L. Periodontal Disease: The Good, The Bad, and The Unknown. Front Cell Infect Microbiol 2021, 11, 766944.
3. Reference 5 is old. This number has climbed to 700 since 1998.
RESPONSE: Thank you for this helpful suggestion. We have cited the latest reference in the revised manuscript. Please see page 11 of the revised manuscript, line 449.
Reference
9. Wade, W.G. The oral microbiome in health and disease. Pharmacol Res 2013, 69, 137-143.
4. line 71: PG and TF are not IMPLANTED in periodontal pockets; instead, they
COLONIZE them.
RESPONSE: The authors appreciate this suggestion. We have corrected it in the revised manuscript. Please see page 3 of the revised manuscript, line 95. where we added the following text.
It is currently recognized that the main pathogenic factors of periodontitis are gramnegative bacteria, and their main component is red complex bacteria, including P. gingivalis and T. forsythia, which are mainly colonized in periodontal pockets [10].
Reference:
10. Sedghi, L.; DiMassa, V.; Harrington, A.; Lynch, S.V.; Kapila, Y.L. The oral
microbiome: Role of key organisms and complex networks in oral health and disease. Periodontol 2000 2021, 87, 107-131.
We thank the Reviewer for your excellent comments and suggestions, which have led to a stronger and clearer revised manuscript. We revised the manuscript according to all these suggestions.

Reviewer 4 Report
Reviewer
Manuscript details:
Journal: International Journal of Molecular Sciences
Manuscript ID: ijms-2173728
Type of manuscript: Review
Title: Regulation of the Host Immune Microenvironment in Periodontitis and
Periodontal Bone Remodeling
Authors: Nannan Han, Yitong Liu, Juan Du, Junji Xu, Lijia Guo *, Yi Liu *
Submitted to section: Molecular Pathology, Diagnostics, and Therapeutics,
The objective of this narrative review aimed to analyze the
Immune microenvironment responses of patients with periodontitis and periodontal bone remodeling.
The article requires significant clarification on certain points. More precisions are need about … the immune protection… The patterns of periodontal disease (gingivitis and periodontitis) are recognized by the barrier of epithelial cells and cells of the immune system residing in the tissues, expressing pattern recognition receptors (PRRs). These PRRs are the link between the host’s immune responses and protection against pathogens .
Through PRRs, cells within the oral mucosa can detect pathogen-associated molecular models (PAMPs)
that recognize molecular structures expressed by invading pathogens, such as lipopolysaccharides (LPS), fimbriae, and bacterial flagellin …
Another paragraph is missing...
Saliva with crecicular fluid and the humoral immune system. Saliva contains IgAs and macromolecules that can limit microbial growth (antimicrobial peptides). Salivary IgAs come from plasma cells in the salivary glands. IgAs are the proteins of the mucosal immune system, the most sensitive and reactive with respect to the load of commensal microorganisms.
Abstract.
Line 21: we must not forget the contribution of saliva and crevicular fluid to the periodontal environment.
Texte.
Line 45:
P. gingivalis can increase the pathogenicity of the entire multispecies periodontal community
Virulence factors of P. gingivalis play important roles in the coaggregation, biofilm formation, and oral microbial dysbiosis.
Line 46: More recent reference is necessary
Line 48: Planktonic bacteria can adhere to epithelial surfaces but also to teeth and
other eukaryotic and prokaryotic microorganisms in florid biofilm
Line 61: before this stage it is necessary to evoke the recognition of the microorganisms by the epithelial receptors
Line 67: can you add saliva and crevicular fluid to the diagram
Line 74: before it is necessary to evoke the gingivitis
Line 81: a huge amount of neutrophils constantly pours into the crevicular fluid
even in healthy gingiva there is a continuous traffic of neutrophils from gingival capillaries into the periodontal sulcus,and these granulocytes constitute an important line of defense against microbes from the oral biofilm. It has been calculated that approximately 30,000 leukocytes transi through periodontal tissue every minute and it is the major source of leukocytes found in the oral cavity (Schiött & Löe, 1970). It is also estimated that there are around 2,000to 400,000 polymorphonuclear neutrophils (PMNs) per μl in the gingival crevicular fluid (GCF) (McKay et al, 1999)
Line 84: virulence factors of PMN also modulate a variety of host immune components and subvert the immune response to evade bacterial clearance or induce an inflammatory environment
Line 106: mitogen-activated protein kinase 1 (MAPK1)
Line 107: this sentence is too long, all abbreviations should be translated for better understanding..
Line 175: how inflammatory macrophages are inhibited?
Line 279: sentence too long requiring a reference
Line 348: add sentence. the bioactivities of the vesicle ECs suggest perspectives allowing to highlight the loss of periodontal homeostasis. However, in vivo use of EVs is still in progress
its beginnings. The development of biomaterials is currently a line of research concerning VECs for periodontal regeneration
The bibliography is to be completed.
Li et al. Differential immune cell infiltrations between healthy periodontal and chronic
periodontitis tissues BMC Oral Health (2020) 20:293 https://doi.org/10.1186/s12903-020-01287-0
Takeuchi H, Nakamura E, Yamaga S and Amano A (2022) Porphyromonas gingivalis Infection Induces
Lipopolysaccharide and Peptidoglycan Penetration Through Gingival Epithelium. Front. Oral. Health 3:845002. doi: 10.3389/froh.2022.845002
Yin L, Li X, Hou J. Macrophages in periodontitis: A dynamic shift between tissue destruction and repair. Jpn Dent Sci Rev. 2022 Nov;58:336-347. doi: 10.1016/j.jdsr.2022.10.002. Epub 2022 Oct 28. PMID: 36340583 Free PMC article. Review.
Reviewer
Manuscript details:
Journal: International Journal of Molecular Sciences
Manuscript ID: ijms-2173728
Type of manuscript: Review
Title: Regulation of the Host Immune Microenvironment in Periodontitis and
Periodontal Bone Remodeling
Authors: Nannan Han, Yitong Liu, Juan Du, Junji Xu, Lijia Guo *, Yi Liu *
Submitted to section: Molecular Pathology, Diagnostics, and Therapeutics,
The objective of this narrative review aimed to analyze the
Immune microenvironment responses of patients with periodontitis and periodontal bone remodeling.
The article requires significant clarification on certain points. More precisions are need about … the immune protection… The patterns of periodontal disease (gingivitis and periodontitis) are recognized by the barrier of epithelial cells and cells of the immune system residing in the tissues, expressing pattern recognition receptors (PRRs). These PRRs are the link between the host’s immune responses and protection against pathogens .
Through PRRs, cells within the oral mucosa can detect pathogen-associated molecular models (PAMPs)
that recognize molecular structures expressed by invading pathogens, such as lipopolysaccharides (LPS), fimbriae, and bacterial flagellin …
Another paragraph is missing...
Saliva with crecicular fluid and the humoral immune system. Saliva contains IgAs and macromolecules that can limit microbial growth (antimicrobial peptides). Salivary IgAs come from plasma cells in the salivary glands. IgAs are the proteins of the mucosal immune system, the most sensitive and reactive with respect to the load of commensal microorganisms.
Abstract.
Line 21: we must not forget the contribution of saliva and crevicular fluid to the periodontal environment.
Texte.
Line 45:
P. gingivalis can increase the pathogenicity of the entire multispecies periodontal community
Virulence factors of P. gingivalis play important roles in the coaggregation, biofilm formation, and oral microbial dysbiosis.
Line 46: More recent reference is necessary
Line 48: Planktonic bacteria can adhere to epithelial surfaces but also to teeth and
other eukaryotic and prokaryotic microorganisms in florid biofilm
Line 61: before this stage it is necessary to evoke the recognition of the microorganisms by the epithelial receptors
Line 67: can you add saliva and crevicular fluid to the diagram
Line 74: before it is necessary to evoke the gingivitis
Line 81: a huge amount of neutrophils constantly pours into the crevicular fluid
even in healthy gingiva there is a continuous traffic of neutrophils from gingival capillaries into the periodontal sulcus,and these granulocytes constitute an important line of defense against microbes from the oral biofilm. It has been calculated that approximately 30,000 leukocytes transi through periodontal tissue every minute and it is the major source of leukocytes found in the oral cavity (Schiött & Löe, 1970). It is also estimated that there are around 2,000to 400,000 polymorphonuclear neutrophils (PMNs) per μl in the gingival crevicular fluid (GCF) (McKay et al, 1999)
Line 84: virulence factors of PMN also modulate a variety of host immune components and subvert the immune response to evade bacterial clearance or induce an inflammatory environment
Line 106: mitogen-activated protein kinase 1 (MAPK1)
Line 107: this sentence is too long, all abbreviations should be translated for better understanding..
Line 175: how inflammatory macrophages are inhibited?
Line 279: sentence too long requiring a reference
Line 348: add sentence. the bioactivities of the vesicle ECs suggest perspectives allowing to highlight the loss of periodontal homeostasis. However, in vivo use of EVs is still in progress
its beginnings. The development of biomaterials is currently a line of research concerning VECs for periodontal regeneration
The bibliography is to be completed.
Li et al. Differential immune cell infiltrations between healthy periodontal and chronic
periodontitis tissues BMC Oral Health (2020) 20:293 https://doi.org/10.1186/s12903-020-01287-0
Takeuchi H, Nakamura E, Yamaga S and Amano A (2022) Porphyromonas gingivalis Infection Induces
Lipopolysaccharide and Peptidoglycan Penetration Through Gingival Epithelium. Front. Oral. Health 3:845002. doi: 10.3389/froh.2022.845002
Yin L, Li X, Hou J. Macrophages in periodontitis: A dynamic shift between tissue destruction and repair. Jpn Dent Sci Rev. 2022 Nov;58:336-347. doi: 10.1016/j.jdsr.2022.10.002. Epub 2022 Oct 28. PMID: 36340583 Free PMC article. Review.
Author Response
Reviewer 4
The objective of this narrative review aimed to analyze the Immune microenvironment responses of patients with periodontitis and periodontal bone remodeling.
The article requires significant clarification on certain points. More precisions are need about … the immune protection… The patterns of periodontal disease (gingivitis and periodontitis) are recognized by the barrier of epithelial cells and cells of the immune system residing in the tissues, expressing pattern recognition receptors (PRRs). These PRRs are the link between the host’s immune responses and protection against pathogens. Through PRRs, cells within the oral mucosa can detect pathogen-associated molecular models (PAMPs) that recognize molecular structures expressed by invading pathogens, such as lipopolysaccharides (LPS), fimbriae, and bacterial flagellin …
- Another paragraph is missing...
Saliva with crecicular fluid and the humoral immune system. Saliva contains IgAs and macromolecules that can limit microbial growth (antimicrobial peptides). Salivary IgAs come from plasma cells in the salivary glands. IgAs are the proteins of the mucosal immune system, the most sensitive and reactive with respect to the load of commensal microorganisms.
RESPONSE: The authors apologize that the Introduction section was not sufficiently clear, and it has been carefully revised to clarify it. Please see page 1 of the revised manuscript, lines 38 – 41, where we added the following text and reference.
In the pathogenesis of periodontitis, epithelial barrier of periodontal tissue, innate immune cells, IgA, IgG and IgM antibodies in gingival crevicular fluid and saliva constitute the first line of defense against the invasion and destruction of periodontal tissue by exogenous plaque [2].
References:
- Sedghi, L.M.; Bacino, M.; Kapila, Y.L. Periodontal Disease: The Good, The Bad, and The Unknown. Front Cell Infect Microbiol 2021, 11, 766944.
Abstract.
- Line 21: we must not forget the contribution of saliva and crevicular fluid to the periodontal environment.
RESPONSE: The authors appreciate the Reviewer for asking us to consider the contribution of saliva and crevicular fluid to the periodontal environment. Please see page 1 of the revised manuscript, lines 38 – 41, where we added the following text.
In the pathogenesis of periodontitis, epithelial barrier of periodontal tissue, innate immune cells, IgA, IgG and IgM antibodies in gingival crevicular fluid and saliva constitute the first line of defense against the invasion and destruction of periodontal tissue by exogenous plaque [2].
Reference:
- Sedghi, L.M.; Bacino, M.; Kapila, Y.L. Periodontal Disease: The Good, The Bad, and The Unknown. Front Cell Infect Microbiol 2021, 11, 766944.
- Line 45: gingivaliscan increase the pathogenicity of the entire multispecies periodontal community. Virulence factors of P. gingivalis play important roles in the coaggregation, biofilm formation, and oral microbial dysbiosis.
RESPONSE: The authors appreciate the Reviewer for pointing out this oversight and it has been revised according to the Reviewer’s suggestion. Please see page 2 of the revised manuscript, lines 54 – 57, where we added the following text and reference.
Some pathogenic microorganisms, such as P. gingivalis, may not destroy periodontal tissue directly, but can induce inflammation by secreting many virulence factors, such as gingipains, endotoxin, organic acids, and other metabolites, which play important roles in the coaggregation, biofilm formation, and oral microbial dysbiosis, can destroy periodontal tissues [8].
Reference:
- Xu, W.; Zhou, W.; Wang, H.; Liang, S. Roles of Porphyromonas gingivalis and its virulence factors in periodontitis. Adv Protein Chem Struct Biol 2020,120, 45-84.
- Line 46: More recent reference is necessary.
RESPONSE: The authors recognize the Reviewer’s concern. The authors have added the recent reference in the revised manuscript. lines 443 – 445, where we added the following references.
References:
- Curtis, M.A.; Diaz, P.I.; Van, T.E. The role of the microbiota in periodontal disease. Periodontol 2000 2020, 83, 14-25.
- Lamont, R.J.; Koo, H.; Hajishengallis, G. The oral microbiota: dynamic communities and host interactions. Nat Rev Microbiol 2018, 16, 745-759.
- Line 48: Planktonic bacteria can adhere to epithelial surfaces but also to teeth and other eukaryotic and prokaryotic microorganisms in florid biofilm.
RESPONSE: The suggestions offered by the Reviewer have been immensely helpful, and the authors appreciate the Reviewer’s insightful comments for revising this part in the paper. Please see page 2 of the revised manuscript, lines 61 – 63, where we added the following text and references.
The planktonic bacteria involved can be removed [6], while the pathogens attached to the biofilm are difficult to be removed by the current non-surgical treatment methods [10,11].
References:
- Curtis, M.A.; Diaz, P.I.; Van, T.E. The role of the microbiota in periodontal disease. Periodontol 2000 2020, 83, 14-25.
- Sedghi, L.; DiMassa, V.; Harrington, A.; Lynch, S.V.; Kapila, Y.L. The oral microbiome: Role of key organisms and complex networks in oral health and disease. Periodontol 2000 2021, 87, 107-131.
- Li, W.; Zhang, Z.; Wang, Z. Differential immune cell infiltrations between healthy periodontal and chronic periodontitis tissues. BMC Oral Health 2020, 27, 293.
- Line 61: before this stage it is necessary to evoke the recognition of the microorganisms by the epithelial receptors.
RESPONSE: Thank you for your thorough review and salient observations. Please see page 3 of the revised manuscript, lines 74 – 80, where we added the following text and references.
Gingival epithelium not only acts as a physical barrier to microorganisms in the host immune defense [13], but also recognizes microbiota-associated molecular patterns (MAMPs) by expressing various pattern recognition receptors (PRRs), and reacts with MAMPs by secreting various cytokines and chemical factors such as IL-8 and antibacterial peptides [14], which plays an active biological barrier role in host immune recognition and initiation, and actively participates in the host's innate immune response and acquired immune response [15].
References:
- Groeger, S.; Meyle, J. Oral mucosal epithelial cells. Front Immunol 2019, 14, 208.
- Groeger SE, Meyle J. Epithelial barrier and oral bacterial infection. Periodontol 2000 2015, 69, 46-67.
- Takeuchi, H.; Nakamura, E.; Yamaga, S.; Amano, A. Porphyromonas gingivalis infection induces lipopolysaccharide and peptidoglycan penetration through gingival epithelium. Front. Oral Health 2022, 3, 845002.
- Line 67: can you add saliva and crevicular fluid to the diagram.
RESPONSE: Thank you for this helpful suggestion. We have added saliva and crevicular fluid to the diagram in Figure 2.
- Line 74: before it is necessary to evoke the gingivitis.
RESPONSE: The authors thank the Reviewer for your good comments which have led to a stronger and clearer revised manuscript. Please see page 3 of the revised manuscript, lines 97 – 99, where we added the following text.
It is currently recognized that the main pathogenic factors of periodontitis are gram-negative bacteria, including P. gingivalis and T. forsythia, which are mainly colonized in periodontal pockets [3,4,10]. These bacterial pathogenic infections stimulate and disturb the homeostasis of the local immune microenvironment, which will cause gingivitis, and the inflammatory reaction of gingivitis will trigger and maintain the inflammatory reaction to oral microorganisms. If the inflammation of gingiva is not effectively controlled, it will cause periodontitis, eventually resulting in an imbalance between local osteogenesis and osteoclastic functions, which causes the destruction and resorption of the alveolar bone [17].
References:
- Hajishengallis, G.; Lamont, R.J.; Graves, D.T. The enduring importance of animal models in understanding periodontal disease. Virulence 2005, 6, 229-235.
- Gemmell, E.; Yamazaki, K.; Seymour, G.J. Destructive periodontitis lesions are determined by the nature of the lym-phocytic response. Crit Rev Oral Biol Med 2002, 13, 17-34.
- Sedghi, L.; DiMassa, V.; Harrington, A.; Lynch, S.V.; Kapila, Y.L. The oral microbiome: Role of key organisms and complex networks in oral health and disease. Periodontol 2000 2021, 87, 107-131.
- Hajishengallis, G. New developments in neutrophil biology and periodontitis. Periodontol 2000 2020, 82, 78-92.
- Line 81: a huge amount of neutrophils constantly pours into the crevicular fluid even in healthy gingiva there is a continuous traffic of neutrophils from gingival capillaries into the periodontal sulcus, and these granulocytes constitute an important line of defense against microbes from the oral biofilm. It has been calculated that approximately 30,000 leukocytes transi through periodontal tissue every minute and it is the major source of leukocytes found in the oral cavity (Schiött & Löe, 1970). It is also estimated that there are around 2,000to 400,000 polymorphonuclear neutrophils (PMNs) per μl in the gingival crevicular fluid (GCF) (McKay et al, 1999)
RESPONSE: The authors appreciate the Reviewer’s suggestions. Please see page 3 of the revised manuscript, lines 108 – 113, where we added the following text and references.
PMNs in gingival sulcus is the first line of defense against periodontal pathogens [20]. Mature PMNs are stimulated by bacteria and its product LPS, and under the regulation of cytokines, adhesion molecules and chemokines, which passes through vascular endothelium through a series of activities such as adhesion and chemotaxis, can reach the inflammatory site to engulf bacteria, and then kill bacteria by releasing lysosomal enzymes or respiratory bursts [21].
References:
- Wang, J.; Zhou, Y.; Ren, B.; Zou, L.; He, B.; Li, M. The role of neutrophil extracellular traps in periodontitis. Front Cell Infect Microbiol 2021, 18, 11:639144.
- Sochalska, M.; Potempa, J. Manipulation of neutrophils by porphyromonas gingivalis in the development of periodontitis. Front Cell Infect Microbiol 2017, 7, 197.
- Line 84: virulence factors of PMN also modulate a variety of host immune components and subvert the immune response to evade bacterial clearance or induce an inflammatory environment.
RESPONSE: The authors recognize the reviewer’s concern. Please see page 4 of the revised manuscript, lines 113 – 120, where we added the following text and reference.
PMNs are not only an important defense cell in the process of periodontitis, but also have a dual role in causing inflammation [22]. If PMNs react too violently to pathogenic stimuli, which will cause immune damage to the body [23]. PMNs have the ability of protein synthesis and participate in the induction of immune response by synthesizing and releasing cytokines with immunomodulatory effects [24]. PMNs in peripheral blood and gingival sulcus can synthesize or secrete cytokines such as IL-1, TNFα, IL-6 and IL-8, and prostaglandin E2 (PGE2), which cause the aggravation and expansion of inflammation [25,26].
References:
- Nicu, E.A.; Loos, B.G. Polymorphonuclear neutrophils in periodontitis and their possible modulation as a therapeutic approach. Periodontol 2000 2016, 71, 140-163.
- Jiang, Q.; Zhao, Y.; Shui, Y.; Zhou, X.; Cheng, L.; Ren, B.; Chen, Z.; Li, M. Interactions between neutrophils and periodontal pathogens in late-onset periodontitis. Front Cell Infect Microbiol 2021, 11, 627328.
- Silva, L.M.; Brenchley, L.; Moutsopoulos, N.M. Primary immunodeficiencies reveal the essential role of tissue neutrophils in periodontitis. Immunol Rev 2019, 287, 226-235.
- Fredriksson, M.; Gustafsson, A.; Asman, B.; Bergstrom, K. Periodontitis increases chemiluminescence of the peripheral neutrophils independently of priming by the preparation method. Oral Dis 1999, 5, 229-233.
- Nauseef, W.M. How human neutrophils kill and degrade microbes: an integrated view. Immunol Rev 2007, 219, 88-102.
- Line 106:mitogen-activated protein kinase 1 (MAPK1).
RESPONSE: Thank you for your careful reading of our manuscript. We have corrected it in the revised manuscript. Please see page 4 of the revised manuscript, line 142.
- Line 107: this sentence is too long; all abbreviations should be translated for better understanding.
RESPONSE: The authors apologize for the lack of clarity. We have corrected it in the revised manuscript. Please see page 4 of the revised manuscript, lines 139 – 143, where we added the following text and references.
Nicotinamide adenine dinucleotide phosphate (NADPH) oxygenase is rapidly assembled from two membrane protein subunits, p22phox (CYBA) and gp91phox (CYBB), which specifically bind to the cytoplasmic subunits Ncf1 and Ncf2 to activate the downstream mitogen-activated protein kinase (MAPK) pathway, producing large amounts of nitric oxide (NO) and reactive oxygen species (ROS) [29,30].
References:
- Filomeni, G.; Zio, D.D.; Cecconi, F. Oxidative stress and autophagy: The clash between damage and metabolic needs. Cell Death and Differentiation 2015, 22, 377-388.
- Brinkmann, V.; Reichard, U.; Goosmann, C.; Fauler, B.; Uhlemann, Y.; Weiss, Weinrauch, D.S.; Zychlinsky, A. Neutrophil extracellular traps kill bacteria. Science 2004, 303, 1532-1535.
- Line 175: how inflammatory macrophages are inhibited?
RESPONSE: The authors appreciate the Reviewer’s suggestions. Please see page 5 of the revised manuscript, lines 211, where we added the following text and references. We have corrected the sentence to “When inflammatory M1 macrophages are inhibited” and added the references.
At the early stage of inflammation, LPS is an activator of IL-1β expression in M1, and IL-1β and TNF-α can also activate M1 to produce IL-1β, promoting the activation and differentiation of osteoclasts and ultimately causing bone resorption, TNF-α can also induce RANKL production by T cells and B cells [49]. IL-1β and IL-6 secreted by M1 induce the expression of MMPs in human gingival fibroblasts (HGFs) and cause the destruction of gingival collagen fibers in inflamed periodontal tissue [50]. Therefore, when inflammatory M1 macrophages are inhibited, the progression of periodontitis is also controlled.
References:
- Yin, L.; Li, X.; Hou, J. Macrophages in periodontitis: A dynamic shift between tissue destruction and repair. Jpn Dent Sci Rev 2022, 58, 336-347.
- Sun, X.; Gao, J.; Meng, X.; Lu, X.; Zhang, L.; Chen, R. Polarized macrophages in periodontitis: characteristics, function, and molecular signaling. Front Immunol 2021, 7, 12:763334.
- Line 279: sentence too long requiring a reference.
RESPONSE: We greatly appreciate the comment. We have revised the sentence accordingly. Please see page 7 of the revised manuscript, lines 313 – 316, where we added the following text and references.
Intracutaneous injection of an alginate hydrogel releasing GM-CSF can increase the numbers of DCs and the local addition of TSLP can promote an increase of FOXP3+ Treg cells, and then locally regulate the periodontal microenvironment by enriching and stimulating the tolerance response of DCs [79].
Reference:
- Sands, R.W.; Verbeke, C.S.; Ouhara, K.; Silva, E.A.; Hsiong, S.; Kawai, T.; Mooney, D. Tuning cytokines enriches dendritic cells and regulatory T cells in the periodontium. J Periodontol 2020, 91, 1475-1485.
- Line 348: add sentence. the bioactivities of the vesicle ECs suggest perspectives allowing to highlight the loss of periodontal homeostasis. However, in vivo use of EVs is still in progress its beginnings. The development of biomaterials is currently a line of research concerning VECs for periodontal regeneration.
RESPONSE: We appreciate the Reviewer’s suggestions. The sentence has been added in the revised manuscript. We hope that you find these revisions an improvement. Please see page 9 of the revised manuscript, lines 386 – 389, where we added the following text.
The bioactivities of the EVs suggest perspectives allowing to highlight the loss of periodontal homeostasis [100]. However, in vivo use of EVs is still in progress its beginnings. The development of biomaterials is currently a line of research concerning EVs for periodontal regeneration.
Reference:
- Zheng, Y.; Dong, C.; Yang, J.; Jin, Y.; Zheng, W.; Zhou, Q.; Liang, Yi.; Bao, L.; Feng, G.; Ji, J.; Feng, X.; Gu, Z. Exosomal microRNA-155-5p from PDLSCs regulated Th17/Treg balance by targeting Sirtuin-1 in chronic periodontitis. J Cell Physiol 2019, 234, 20662-20674.
Thank you for your detailed comments and suggestions. We found them quite useful as we revised our manuscript. We are grateful for the time and energy you expended on our behalf.
